# Recent Advances in Antimicrobial Nano-Drug Delivery Systems

**DOI:** 10.3390/nano12111855

**Published:** 2022-05-29

**Authors:** Tong-Xin Zong, Ariane Pandolfo Silveira, José Athayde Vasconcelos Morais, Marina Carvalho Sampaio, Luis Alexandre Muehlmann, Juan Zhang, Cheng-Shi Jiang, Shan-Kui Liu

**Affiliations:** 1School of Biological Science and Technology, University of Jinan, Jinan 250022, China; 202021201243@stu.ujn.edu.cn (T.-X.Z.); zjandzq@163.com (J.Z.); 2Institute of Biological Sciences, University of Brasília, Brasilia 70910900, Brazil; pandolfo.ariane@gmail.com (A.P.S.); joseavmorais@gmail.com (J.A.V.M.); marinacsampaio@gmail.com (M.C.S.); 3Faculty of Ceilandia, University of Brasilia, Brasilia 72220900, Brazil

**Keywords:** antimicrobial, antimicrobial resistance, drug delivery systems, nanoparticles, nanocarriers

## Abstract

Infectious diseases are among the major health issues of the 21st century. The substantial use of antibiotics over the years has contributed to the dissemination of multidrug resistant bacteria. According to a recent report by the World Health Organization, antibacterial (ATB) drug resistance has been one of the biggest challenges, as well as the development of effective long-term ATBs. Since pathogens quickly adapt and evolve through several strategies, regular ATBs usually may result in temporary or noneffective treatments. Therefore, the demand for new therapies methods, such as nano-drug delivery systems (NDDS), has aroused huge interest due to its potentialities to improve the drug bioavailability and targeting efficiency, including liposomes, nanoemulsions, solid lipid nanoparticles, polymeric nanoparticles, metal nanoparticles, and others. Given the relevance of this subject, this review aims to summarize the progress of recent research in antibacterial therapeutic drugs supported by nanobiotechnological tools.

## 1. Introduction

Bacteria can cause life-threatening human diseases, and especially superbug infections kill at least 700,000 people worldwide every year [1]. Antibacterial drugs are indispensable drugs for the treatment of bacterial infections. However, due to the excessive use of antibiotics in clinical, agricultural, and animal production environments, microorganisms have become increasingly resistant to antibiotics, which can lead to a variety of bacterial infections, and seriously threaten human health and life.

Antimicrobial resistance is the ability of a microorganism to escape the mechanism of action of antimicrobial drugs, and thus survive [2]. In principle, most ATBs need to pass through the microbial cell envelope (for gram-negative bacteria, at least through the outer membrane) to reach their target site, as shown in Figure 1. As a result, ATB can disrupt the biochemical processes in cells, thereby inhibiting cell growth (bacteriostatic drugs) or leading to cell death (bactericidal drugs) [3]. However, pathogens have rapidly developed various ways to evade the toxic effects of ATBs. For example, the restriction of access of the ATB to its target site by preventing entry or increasing efflux, the modification and protection of the ATB targets, and inactivation of ATB drugs are to be cited as frequent resistance mechanisms [4,5].

Certain antibiotics have intracellular targets and thus must at least enter the bacterial cell through the pore protein (a) through the outer membrane, and through the inner membrane through the active transport channel (a, b); once the antibiotic enters the bacterial cell, it binds to its target site, disrupt biochemical processes of bacteria, leading to cell growth inhibition or cell death. In the case of antimicrobial resistance, bacteria may develop different evasion strategies, such as reducing the number of pores and transport channels, or changing the structure and function of these membrane proteins to prevent the entry of the antibiotic (b); producing enzymes to destroy or change the chemical structure of ATB molecules in the periplasm or cytoplasm (c, d, e); generating alternative binding sites to prevent vulnerable therapeutic targets (t) from binding to the antibiotic (f); changing the structure and function of the target (f); and developing or upgrading a complex efflux pump (c) to quickly pump the antibiotic out of bacterial cells (h).

Antibiotic resistance is regarded as one of the greatest health threats by the World Health Organization (WHO) [6,7]. So far, it has become a global concern and threatened the effectiveness of basic infection prevention and treatment worldwide [8]. Restricting the use of antibiotics while improving sanitation and antibiotic disposal measures has always been the main action plan proposed by the World Health Organization (WHO) and government health agencies [9]. Therefore, we need more international cooperation and multidisciplinary research projects to protect and strengthen public health strategies against the problem of antimicrobial resistance around the world [10]. However, developing new antibiotics, biological products or adjuvant therapies is not only a difficult process, but also cannot keep up with the growing rate of drug resistance. Thus, it urgently needs new clinically approved antimicrobial therapies [11]. Nano-formulations have recently become a way to combat antibacterial resistance and overcome the challenges associated with delivery of antibacterial agents, such as low bioavailability, sub-therapeutic drug accumulation in microbial reservoirs, drug-related toxicities, and frequent drug dosing regimen [12]. As a result, the combination of antibiotics and nanotechnology has increasingly attracted attention of pharmaceutical scientists. Due to the significance of nanodrugs, this review herein will summarize the recent developments of diverse nanosystems as antimicrobial drug delivery agents in overcoming bacterial resistance, and their action mechanisms as well.

## 2. Antimicrobial Nanomedicine Drug Delivery Systems

The development of nano-drug delivery systems (NDDSs) provides new methods for the prevention and treatment of diseases. In recent years, liposomes, polymeric micelles, nanogels, inorganic nanoparticles, and inorganic/organic (core/shell) nanoparticles have been widely used as NDDS in pre-clinical applications and clinical research [13]. These systems can improve the pharmacological and pharmaceutical properties of drugs by extending the time of action, improving the efficacy and targeting, overcoming drug resistance, and reducing immunogenicity and toxicity [14,15]. Nowadays, more and more NDDS are being applying in the preclinical and clinical phases [16].

### 2.1. Liposomes

Liposomes can be listed as one of the most promising NDDS used in the field of antibiotics under study [17,18]. These lipid-based nanosystems were initially developed as drug carriers in the 1970s. Since then, major breakthroughs in liposome technology have led them to be highly effective antimicrobial DDS [19,20,21]. Structurally, liposomes are spheroid vesicles constituted by an aqueous core surrounded by one or multiple lipid bilayers, usually composed of amphiphilic lipids, such as phospholipids. The size of the liposomes commonly ranges from 20 nm to tens of microns, and the thickness of each membrane is about 4 nm. Liposomes have many advantages compared with other DDSs, by showing their unique potential to combine hydrophilic (Figure 2a) and hydrophobic (Figure 2b) small-molecular drugs in a single structure, biocompatibility, biodegradability, low toxicity, and low interaction with the immune system [20,22,23]. It has been found that cationic liposomes, anionic liposomes and neutral liposomes can be obtained by effectively charging charges on the surface of liposomes (Figure 2c). The main forms of interaction between liposomes and cells include endocytosis (Figure 2d) and fusion (Figure 2e). In addition, liposomes can be easily coupled with other targeting macromolecules (e.g., antibodies, proteins, or enzymes), so that drugs can be effectively delivered to the site of infection (Figure 2g) [24,25]. More importantly, several liposome preparations represented by Arikayce liposome for the treatment of Mycobacterium avium had been approved by FDA or showed great therapeutic potential in clinical trials of anti-infectious diseases. For example, Muppidi et al. had used pegylated liposomes to encapsulate vancomycin, and the results showed that the pegylated liposome system could increase the concentration of vancomycin in body, so that methicillin-resistant *Staphylococcus aureus* (*MRSA*) infection pneumonia can be effectively treated with the risk of drug nephrotoxicity reduced [26]. Since pegylation could overcome the problem that traditional liposomes are easily taken up by the reticuloendothelial system by significantly increasing the circulation time of the delivery system in the body, the modification of polyethylene glycol had made the liposomal drug delivery system a successful step in extending the circulation time in the body and increasing the half-life (f) [27,28].

Therefore, liposomes can also avoid the drug resistance mechanism of bacteria due to changes in membrane permeability and the production of hydrolytic enzymes. After the liposomes reach the bacteria, they can interact with the lipoproteins and lipopolysaccharides of the bacterial outer membrane, fuse with its cell membrane, and then release the antibacterial drugs to inhibit bacterial proliferation. Mohammad Darvishi et al. [29] prepared liposomal moxifloxacin formulations by thin film hydration, the bilayer is composed of cholesterol and phospholipids in a 30:70 molar ratio. To prepare cationic liposomes, a 5% cationic agent (CTAB) was added. The Minimum Inhibitory Concentration (MIC) of free drug, neutral moxifloxacin and cationic moxifloxacin liposomes against *Pseudomonas aeruginosa* were 10, 5 and 2.5, respectively. The MICs of free drug, neutral moxifloxacin and cationic moxifloxacin liposomes against *S. aureus* were 1, 1 and 0.5, respectively. This study shows that encapsulation of moxifloxacin into liposomes, especially cationic vesicles, can enhance antibacterial properties.

Meers et al. [30] studied the inhibitory effect of liposome formulations of amikacin (Arikace^®^) on biofilms of *Pseudomonas aeruginosa*. Comparing that the free amikacin treatment did not change the colony forming unit (CFU) counts, the liposomal amikacin reduced CFU by about 40%. The results showed that the monolayer cationic particles were smaller, and thus entered the biofilms of *Pseudomonas aeruginosa* and *Staphylococcus aureus* faster than larger particles. It was speculated that the unilamellar cationic liposomes might change the electrostatic balance of the colony, so that the antibiofilm agent can play a more effective role [31]. In a separate study, the effect of neutral and anionic liposomes tobramycin has also been tested, but the effect of liposomes tobramycin was not better than that of its free drug [32]. It was obvious that the electrical properties of liposome played an important role in the function of drugs.

Lipid-based liquid crystalline nanoparticles (LCNPs) formed by self-emulsification of a certain concentration of amphiphilic lipid materials and surfactants in water are promising candidates for drug delivery applications. LCNPs possess closed lipid bilayers and honeycomb structure. Based on the unique internal dual-channel structure and huge membrane surface area, LCNPs can encapsulate various polar (hydrophilic, lipophilic or amphiphilic) drugs. In addition, LCNPs have certain advantages, including the simple preparation process, biocompatibility, adhesiveness, and biodegradability, and they can solubilize, encapsulate, protect drugs, promote drug absorption, and improve drug stability. For example, Glycoside hydrolase (PSLG) enhanced the activity of antibiotics in bacterial biofilm-related infections by degrading the essential polysaccharide PSL in the biofilm matrix [33,34], and the combination of antibiotics and PSLG was used to eradicate Pseudomonas aeruginosa infection. However, PSLG is sensitive to proteolysis. To stabilize and protect PSLG from degradation by proteases, Chelsea R. Thorn et al. [35] have used LCNPs as delivery systems for PSLG and tobramycin. The results indicated the prepared LCNPs could protect the enzyme from proteolysis, trigger and maintain the release of PSLG, increase the antibacterial effect by 10–100 times. Therefore, the therapeutic effect of the drug should be significantly enhanced by liposomes.

Although liposomes have significant advantages, they also suffer from some limitations, such as difficulty in controlling particle size distribution, micron size and high levels of solvent residues, and low capture efficiency of hydrophilic compounds. In response to these limitations, a supercritical-assisted process called SuperLip (Supercritical-Assisted Liposome Formation) has been proposed in the literature for the stable production of liposomes over a one-year observation period with lower solvent residues than in the U.S. Food and Drug Administration restrictions, particle size distribution is well controlled, and liposome production can be controlled at the nanometer or micrometer level [36]. In addition, the process has been tested for encapsulation of antibiotics, achieving up to 99% encapsulation efficiency [37].

In Sybil Obuobi’ study [38], a hybrid system (Van_DNL) is fabricated wherein nucleic acid nanogels are caged within a liposomal vesicle for antibiotic delivery. The central principle of this approach relies on exploiting non-covalent electrostatic interactions between cationic cargos and polyanionic DNA to immobilize antibiotics and enable precise temporal release against intracellular *S. aureus*. In vitro characterization of Van_DNL revealed a stable homogenous formulation with circular morphology and enhanced vancomycin loading efficiency. The hybrid system significantly sustained the release of vancomycin over 24 h compared to liposomal or nanogel controls. Under enzymatic conditions relevant to *S. aureus* infections, lipase triggered release of vancomycin was observed from the hybrid. While using Van_DNL to treat *S. aureus* infected macrophages, a dose dependent reduction in intracellular bacterial load was observed over 24 h and exposure to Van_DNL for 48 h caused negligible cellular toxicity. Pre-treatment of macrophages with the antimicrobial hybrid resulted in a strong anti-inflammatory activity in synergy with vancomycin following endotoxin stimulation. Conceptually, these findings highlight these hybrids as a unique and universal platform for synergistic antimicrobial and anti-inflammatory therapy against persistent infections.

Traditional methods for liposomes production, such as Bangham technique [39], microfluidic channel [40] and ethanol injection [41], suffer of several drawbacks: it is difficult to control liposomes particle size and distribution, toxic solvent residues, encapsulation efficiencies frequently lower than 30% and discontinuous process layouts [42]. These problems can determine toxicity of liposome formulations, low biocompatibility and difficulties in replicability. Moreover, post-processing steps are often required to obtain smaller liposomes diameter [43]. Liposomes incorporated in hydrogels have emerged as an attractive strategy to reduce undesirable effects such as local accumulation of drugs [44] and consequent toxic responses [45]. Curcumin is a plant chemical with a wide range of biological activities, including anticancer [46], anti-inflammatory [47] and prominent antibacterial activities [48]. High-pressure processing (HPP) is a non-thermal pasteurization technology that can destroy microbial membranes and enzymes in foods via high pressure (commercially at 600 MPa) for the preservation of food nutrition but protect the functional ingredients (flavonoids, protein, and polyphenols). Hua Wei Chen et al. [49] produced curcumin liposomes by a novel combination of ethanol injection and HPP in order to enhance the stability and preservation of curcumin. As a result, the liposomes of curcumin were successfully produced to effectively enhance the stability of curcumin by the novel combination of ethanol injection and HPP method. The novel combination of ethanol injection and HPP technology for the preparation of liposomes is potentially applied in the pharmaceutical, cosmetic, and dairy beverage industries predominantly to reduce the particle size of liposomes, increase their uniform of distribution, inactivate the microorganisms, protect bioactive compounds, and prevent the thermal degradation of bioactive and heat-sensitive compounds.

Liposomes have stable drug loading, prolonged pharmacokinetics, reduced off-target side effects, and improved delivery efficiency to disease targets through strong blood-brain or plasma membrane barriers. At present, a variety of liposome drugs have been listed or are in the clinical research stage, among which the proportion of antibacterial drugs is not small, such as Amikacin Sulfate liposome Injection, amphoericin B liposome for injection, Econazole Nitrate Liposome Gel have been listed. As a drug carrier, liposome is a type of targeted preparation with the earliest clinical application and the most mature development. It was believed that with the development of medical science and the improvement of liposome preparation technology, liposomes will surely bring more surprises to mankind and benefit mankind.

### 2.2. Polymeric Micelles

Polymer micelles are self-assembled structures formed by amphiphilic block copolymers in aqueous solutions (Figure 3). They are solid spherical aggregates with sizes ranging from 10 to 100 nm, composed of amphiphilic copolymers containing hydrophobic and hydrophilic blocks [50]. Similar to small-molecule surfactants, amphiphilic polymer molecules in aqueous solutions exist as single molecules at low concentrations. When the concentration exceeds the critical micelle concentration (CMC), due to hydrophobic, electrostatic, hydrogen bonding, and other molecular forces, the hydrophobic regions of the polymer associate with each other to form micelles. Due to their biophysical and chemical properties, they have shown great potential in DDS [51,52].

An active compound of Beauveria bassiana, 4-Allylpyrocatechol (APC) has a strong inhibitory effect on many oral pathogens, especially *Streptococcus mutans* [53,54]. However, the poor solubility of APC in water limits its application in medicine and clinical. Thus, Siriporn Okonogi et al. [55] prepared APC-loaded polymer micelles (PMAC) by film hydration to improve the water solubility of APC. The result shows that the ratio of APC to polymer has an important influence on the physical and chemical properties of PMAC. When the ratio is 1:4, a smaller particle size (38.8 ± 1.4 nm), a narrower particle size distribution (PDI, 0.28 ± 0.10), high negative zeta potential (−16.43 ± 0.55 mV) and high encapsulation efficiency (86.33 ± 14.27%) of PMAC can be obtained, and the water solubility is significantly improved, about 1,000 times that of unembedded. The transmission electron microscope photograph shows that PMAC is spherical. The inhibitory effect of PMAC (1.5 mg APC/mL) on the biofilm of *Streptococcus intermedius* and *Streptococcus mutans* was significantly stronger than that of chlorhexidine (0.06 mg/mL). Confocal laser confocal microscopy images showed that the biofilm of pathogenic bacteria was destroyed and the thickness decreased after contact with PMAC. The MTT method confirmed that the concentration of PMAC is nontoxic to normal cells. These results indicated that PMAC was a promising natural and harmless antibacterial agent, which is suitable for oral biofilm to inhibit pathogenic bacteria.

Norfloxacin is considered as a poorly soluble drug, despite of the experimental partition coefficient logP that is reported in literature ranging from −0.43 to −1.52, quite different from the value obtained from theoretical calculation (−0.92 to 1.44) [56]. Cremophor EL is FDA-approved nonionic emulsifier, used as a solubilizing agent for many years [57]. Micellar carriers that contain Pluronic polymers exhibit good biocompatibility over a large concentration range, characteristic confirmed in many studies [58]. Maria Antonia Tănase et al. [59] investigated novel polymeric mixed micelles of Pluronic F127 and Cremophor EL as a drug delivery system for norfloxacin as a model antibiotic drug. The drug-loaded mixed micellar formulation exhibit good activity against clinical isolated strains, compared with the CLSI recommended standard strains (*Staphylococcus aureus* ATCC 25923, *Enterococcus faecalis* ATCC 29213, *Pseudomonas aeruginosa*, *Escherichia coli* ATCC 25922). *P. aeruginosa* 5399 clinical strain shows low sensitivity to Norfloxacin in all tested micelle systems. The results suggest that Cremophor EL-Pluronic F127 mixed micelles can be considered as novel controlled release delivery systems for hydrophobic antimicrobial drugs.

One potential approach to address the rising threat of antibiotic resistance is through novel formulations of established drugs. Xingyue Yang et al. [60] designed antibiotic cross-linked micelles (ABC-micelles) by cross-linking the Pluronic F127 block copolymers with an antibiotic itself, via a novel one-pot synthesis in aqueous solution. ABC-micelles enhanced antibiotic encapsulation while also reducing systemic toxicity in mice. Colistin had been used as the “last resort” for clinical multidrug resistant (MDR) bacterial infection management once bacteria have developed resistance mechanisms to almost all other antibiotics [61]. However, the severe side effects of nephrotoxicity [62,63] and neurotoxicity [64] induced by colistin are the bottleneck for its wide use in clinics. The encapsulation efficiency of colistin in ABC-micelles was 80%, and the encapsulated colistin ABC-micelles had good storage stability. ABC-micelles exhibited an improved safety profile, with a maximum tolerated dose of over 100 mg/kg colistin in mice, at least 16 times higher than the free drug. Colistin-induced nephrotoxicity and neurotoxicity were reduced in ABC-micelles by 10–50-fold. Despite reduced toxicity, ABC-micelles preserved bactericidal activity. Rifampicin was known to be one of the most effective antibiotics for intracellular infection treatment [65], but it is not recommended to be used alone due to high risk of resistance [66]. The clinically relevant combination of colistin and rifampicin (co-loaded in the micelles) showed a synergistic antimicrobial effect against antibiotic-resistant strains of *Escherichia coli*, *Pseudomonas aeruginosa*, *and Acinetobacter baumannii*. In a mouse model of sepsis, colistin ABC-micelles showed equivalent efficacy as free colistin but with a substantially higher therapeutic index. Microscopic single-cell imaging of bacteria revealed that ABC-micelles could kill bacteria in a more rapid manner with distinct cell membrane disruption, possibly reflecting a different antimicrobial mechanism from free colistin. This work shows the potential of drug cross-linked micelles as a new class of biomaterials formed from existing antibiotics.

Xingyue Yang et al. previously reported antibiotic-cross-linked micelles, termed ABC-micelles, to reduce the nephrotoxicity and neurotoxicity of colistin; however, only loaded colistin could be released and cross-linked drug as carriers were unable to responsively release cargo in a self-immolation manner [60]. For the targeting intracellular bacteria, this prior work did not address macrophage targeting or cell penetration moieties. To address this, they developed SIR-micelles(+), as a new delivery vehicle comprising antibiotic-loaded micelles with rapid self-immolation within cells for targeted delivery to macrophages, where most intracellular bacterial reside [67]. Colistin was encapsulated in SIR-micelles with 40% yield and good short-term storage stability. Hydrophobic moieties and mannose ligands in SIR-micelles(+) enhanced the delivery of colistin into macrophages. The traceless and thiol-responsive release of colistin effectively eliminated intracellular *Escherichia coli* within twenty minutes. In a murine pneumonia model, SIR-micelles(+) significantly reduced bacterial lung burden of multidrug-resistant Klebsiella pneumoniae. Furthermore, SIR-micelles(+) improved the survival rate and reduced the bacterial burden of organs infected by intracellular bacteria transferred from donor mice. Using this formulation approach, the nephrotoxicity and neurotoxicity induced by antibiotic were reduced by about 5–15 folds. Thus, SIR-micelles(+) represent a new class of material that can be used for targeting treatment of intracellular and drug-resistant pathogens.

Photodynamic antimicrobial therapy (PDAT) is a combination of photosensitizers, light, and molecular oxygen to reactive oxygen species (ROS), such as singlet oxygen (^1^O_2_), superoxide radical anions (O^2−^), hydrogen peroxide (H_2_O_2_) and hydroxyl radicals (HO^.^). It is a clinical treatment method, which has attracted more and more attention because of its precise controllability, high temporal and spatial accuracy, and non-invasiveness [68]. Ren and coworkers [69] successfully prepared a pH-responsive zwitterionic polyurethane nanomicelle. This nano system also has near-infrared bioimaging function, which can detect bacteria in real time. In addition, under normal conditions (pH 7.4), hydrophilic zwitterionic polyurethane plays a vital role in antifouling, improving biocompatibility, and extending cycle time. Under acidic conditions (pH 5.4), the zwitterion part is suddenly protonated, allowing the positively charged nanomicelles to target the bacterial infection site, resulting in a great antibacterial effect, which is considered to be superior to the existing PDAT method.

Piperacillin/tazobactam is considered a safe antimicrobial agent with broad antibacterial activity [70,71,72]. In order to improve the antibacterial properties of piperacillin/tazobactam, a new polymer micelle composed of polyethylene glycol-methyl ether block poly (lactide-glycolide) (PLGA-PEG) supported by piperacillin/tazobactam was developed by Milani Morteza et al. [73]. The result showed that this formulation based on micellar nanocarriers could improve the effectiveness of piperacillin/tazobactam. Compared with the free drug form, piperacillin/tazobactam PLGA-PEG micelles are more effective against resistant *Pseudomonas aeruginosa* [73,74,75] Milani Morteza et al. believed that PLGA-PEG micelles can be used as a good carrier to transport antibiotics into the bacterial biofilm and eliminate the formation of biofilm, probably being an attractive strategy to control the infection of drug-resistant *Pseudomonas aeruginosa* [73].

Lee et al. [76] prepared AMP-covered micelles by the co-assembly of chimeric antimicrobial lipopeptide (DSPE-PEG-HnMc) and a biodegradable amphiphilic polymer (poly-(lactic-co-glycolic acid)-b-poly(ethylene glycol), (PLGA-b-PEG)). The HnMc and PEG formed the shell of the micelles in which PEG protected HnMc from proteolytic degradation. Moreover, HnMc on the surface could help micelles in preferentially binding and killing bacteria. Due to the synergy between HnMc and PEG, the micelles targeted a wide range of bacteria preferentially including *Escherichia coli (E. coli)*, *Listeria monocytogenes*, *Pseudomonas aeruginosa (P. aeruginosa)* and *S. aureus* instead of mammalian cells. Moreover, in vivo experiments also demonstrated superior anti-inflammatory effects of the micelles in a mouse model of drug-resistant *P. aeruginosa* lung infection with highly targeted abilities.

As mentioned above that AgNPs have been one of the promising antibacterial agents. to avoid the aggregation of AgNPs caused by the larger specific surface area and higher surface energy, Lin et al. [77] developed and synthesized a six-arm star polymer micelle composed of PCL, 2-(dimethylamino)ethyl methacrylate (DMAEMA), and polyethylene glycol-methyl ether-methacrylate (PEGMA). Micellar PDMAEMA core is both a reducing agent and a stabilizer, which provides a loading platform for the in situ conversion of precursor silver nitrate into AgNPs. Compared with linear copolymer silver-loaded micelles, the stable AgNPs in star copolymer micelles had smaller average particle size, better anti-dilution and thermal stability, and enhanced antibacterial activity against *E. coli* DH5α due to the serious damage of loaded AgNPs to bacterial membrane. In addition, both types of AgNPs stabilized by copolymer micelles have good cytocompatibility to HepG2 cells.

Curcumin is a plant chemical with a wide range of biological activities, including anticancer [78], anti-inflammatory [79] and prominent antibacterial activities [80]. However, its low solubility and bioavailability make it unlikely to be a clinical drug [81]. To improve the druggability of curcumin, the preparation of micelle nanoparticles has attracted much attention. Biodegradable PLGA-Dex10curc micelles with curcumin as active ingredient were found to enhance their antibacterial and antibiofilm activities due to the functionalization of the dextran shell of the micelles [82]. Due to the interaction between positively charged nanoparticles and negatively charged bacterial cell walls, positively charged nanoparticles usually have strong invasiveness to Gram-negative bacteria. Since PLGA-Dex10curc has a positive charge, it has a significant inhibitory effect on the growth of bacteria. Compared with free curcumin and PLGA-Dex10, the PLGA-Dex10curc nano-drug can effectively destroy the biofilm of Pseudomonas fluorescens at low concentrations, and reduce the cell viability of *Pseudomonas putida* at all concentrations [82].

Multifunctional antimicrobial peptides that combine the intrinsic microbicidal property of cationic polypeptide chains and additional antibacterial strategy hold promising applications for the treatment of infections caused by antibiotic-resistant bacteria, especially “superbugs”. Zheng et al. [83] designed and synthesized a star-shaped copolymer ZnPc-g-PLO with a zinc phthalocyanine (ZnPc) core and four poly(l-ornithine) (PLO) arms as a bifunctional antibacterial agent, i.e., intrinsic membrane damage and photothermal ablation capability. In an aqueous solution, amphiphilic ZnPc-g-PLO molecules self-assemble into nanosized polymeric micelles with an aggregated ZnPc core and star-shaped PLO periphery, where the ZnPc core exhibits appreciable aggregation-induced photothermal conversion efficiency. In the absence of laser irradiation, ZnPc-g-PLO micelles display potent and broad-spectrum antibacterial activities via physical bacterial membrane disruption as a result of the high cationic charge density of the star-shaped PLO. Upon laser irradiation, significant improvement in bactericidal potency was realized due to the efficacious photothermal sterilization from the ZnPc core. Notably, ZnPc-g-PLO micelles did not induce drug resistance upon subinhibitory passages. In summary, dual-functional ZnPc-g-PLO copolymers can serve as promising antibacterial agents for the treatment of infectious diseases caused by antibiotic-resistant bacteria.

Paclitaxel Polymeric Micelles for Injection for cancer treatment have been approved in China, which shows their therapeutic potential and the possibility of rapid approval of subsequent formulations.

### 2.3. Nanogels

Nanogels are nanoscale hydrogels, which are characterized with a polymer network composed of amphiphilic or water-soluble polymers maintained by physical or chemical interactions (Figure 4). The advantages of nanogels, such as easy dispersion in water, having hydrophilic and soft appearance, containing a large amount of water, ability of encapsulating a large number of biologically active molecules (large and small molecules), and good compatibility in the body, lead them to be valuable NDDSs. The diameter of nanogel basic structures is usually between 5 and 500 nm. This effective size range plays an important role in avoiding rapid renal separation, which is sufficient to avoid the uptake of reticuloendothelial system. Due to the nanosize characteristics, nanogels can easily cross the blood-brain barrier (BBB) and show potential permeability [84]. In addition to physical packaging, drugs can also be wrapped in nanogels through salt formation, hydrogen bonding, and hydrophobic interactions [85,86]. The surface of the nanogel is hydrophilic, it is not easy to be opsonized in the blood, and it can prevent macrophages from phagocytosis [87]. Nanogels with different structures and compositions can encapsulate various types of drugs, including hydrophilic drugs, hydrophobic drugs, peptide and protein drugs, and nucleic acid drugs. Nanogels have high encapsulation efficiency, good stability, and environmental sensitivity (such as ionic strength, pH, temperature), and are a very promising system. In addition, nanogels have high biocompatibility, biodegradability and other characteristics, including improving efficacy of drug-loaded chemotherapeutic drugs on drug-sensitive and drug-resistant cancer cells, and the sensitization of refractory bacterial pathogens to antibiotics [88]. Thus, nanogels are currently a very promising NDDS in the pharmaceutical and biotechnology fields [89].

Tuning the composition of antimicrobial nanogels can significantly alter both nanogel cytotoxicity and antibacterial activity. Gu et al.’s project [90] investigated the extent to which PEGylation of cationic, hydrophobic nanogels altered their cytotoxicity and bactericidal activity. The result showed that all tested nanogels decreased the membrane integrity (0.5 mg/mL dose) for Gram-negative *E. coli* and *P. aeruginosa*, irrespective of the extent of PEGylation. Therefore, PEGylation reduced nanogel toxicity to mammalian cells without significantly compromising their bactericidal activity. These results facilitate future nanogel design for perturbing the growth of Gram-negative bacteria.

Zu et al. [91] synthesized poly (*N*-isopropylacrylamide-co-*N*-[3-(dimethylamino)propyl] methacrylamide) (P(NIPAM-co-DMAPMA)-based nanogels by one-pot precipitation polymerization. The antimicrobial agent, triclosan, capsulated in the prepared nanogels achieved in the presence of bacteria when the hydrophobic groups of gels interacted with the lipid membrane. Due to the intercalation and permeation of aliphatic chains on the bacterial cell membrane, the interaction between them and triclosan is no longer so strong, which makes it easier and more direct to release triclosan when the bacterial cell membrane is opened. Therefore, the effective local concentration of triclosan in bacterial parts increased significantly. The minimum bactericidal concentration decreased hundreds of times, significantly improving the therapeutic effect of Triclosan, consequently reducing the use of concentrations without loss of activity. Besides, this nanogels system is expected to reduce the future development of bacterial resistance and reduce the side effects of antibacterial agents.

Azithromycin could inhibit the bacterial movement and the production of several virulence factors (protease, pyocyanin and exopolysaccharide) of Pseudomonas aeruginosa at a dose lower than the minimum inhibitory concentration (MIC) [92,93,94,95]. However, the efficacy of azithromycin in vivo is affected by infection site concentrations, especially in bacterial biofilms [96]. To improve the in vivo potency of Azithromycin, two promising polymer nanoparticle systems, including octenyl succinic anhydride-hyaluronic acid (OSA-HA) nanogels and TPGS-PLGA nanoparticles, were compared and evaluated by Sylvia et al. [97]. The results showed that: (1) Compared with TPGS-PLGA nanoparticles, the interaction between hydrophilic OSA-HA nanogel and mucin was weaker. (2) Both drug delivery systems can penetrate bacterial biofilm, but smaller T-PLGA nanoparticles have longer retention times in biofilm than hydrophilic OSA-HA nanoparticles. (3) Both systems can improve the antibacterial activity of azithromycin, and can prevent and remove bacterial biofilms at lower doses. (4) This nanogel has good cell selectivity and no obvious toxicity to liver cells and lung epithelial cells. Although the two preparations had different characteristics, they both showed good effects on azithromycin mucosal delivery in the treatment of pulmonary infection caused by P. aeruginosa. However, in order to fully evaluate the potential of this formulation, researchers need to develop the final dosage form and in vivo efficacy proof.

Antimicrobial peptides (AMPs) are a new class of drugs with high antibacterial activity, which exist in mammals, plants, fungi and bacteria [98]. Due to their special properties, such as remedial effect [99,100], control of toxicity [101], availability of broad-spectrum and narrow-spectrum peptides, and bioengineering capabilities, they can be used in drug delivery systems. On the other hand, the use of AMPs has some limitations because of its short half-life and fast enzymatic degradation [75]. Nanogels are an appropriate drug delivery route for AMPs because of their high drug loading, biodegradability, adjustable ability, and controlled drug release [102,103]. Chondroitin sulfate is an important pH-sensitive GAG family, which widely exists in extracellular matrix and cell surface of many soft tissues. In addition, due to biocompatibility and appropriate physiological characteristics, it has been used in many fields of biomedical engineering, including drug delivery and tissue engineering [104,105,106]. PLLA can react with other materials through esterification, and it is widely used in in vivo research and clinical application due to its biocompatibility and biodegradability [74,107]. Thus, Sobhan and coworkers prepared poly (L-lactide)-grafted-chondroitin sulfate (PLLA-g-CS) copolymers with different L-lactide contents by ring-opening polymerization [108]. Their study confirmed that the prepared nanogel had potential antibacterial effects and reduced the side effects of nisin on normal HDF cells [108].

Weldrick et al. [75] developed a new type of surface-functionalized tetracycline and lincomycin nanogels to overcome their antibiotic resistance. The nanocarriers were coated by biocompatible cationic polyelectrolyte (bPEI) with mild cross-linking polyacrylic acid nanogel (Carbopol Aqua SF1). The results showed that the bPEI surface-coated nanogels with antibiotics had enhanced effects on several tetracycline- or lincomycin-resistance bacteria. In addition, it was also found that the cytotoxicity of antibiotics loaded on these nanocarriers to human keratinocytes was negligible in the concentration range effective against drug-resistant and sensitive bacteria. This work was believed to provide an important new opportunity to broaden the spectrum and the clinical importance of existing antibiotics by using cheap materials.

In Chen and coworkers’ study, the nanogels with cross-linked cationic poly (L-lysine)-block-poly (L-threonine) (PLL-b-PLT) blockcopeptides were synthesized to carry TRAIL (tumor necrosis factor-related apoptosis-inducing ligand), which was an antibacterial agent for the treatment of sepsis caused by Klebsiella pneumoniae [75]. This nanogel could bind to a bacterial wall by electrostatic interaction, while LPS (lipopolysaccharide) overactivated macrophages phagocytose complex and induce apoptosis due to the release of TRAIL. This result showed that intraperitoneal injection of TRAIL nanogel could prevent LPS-induced lung and kidney injury in mice and intratracheal sepsis induced by Klebsiella pneumoniae [109].

Chlorhexidine gluconate (CHX) salt is a broad-spectrum antibacterial against yeasts, especially for the treatment of oral diseases [27,110,111]. Mohammed et al. [112] associated CHX to nanogel particles based on partially cross-linked acrylate copolymer for antibacterial, antifungal and algae-resistant applications. the stability of CHX-Carbopol nanogel particles (CLC), the encapsulation efficiency of CHX, the controlled release of CHX-cation and the antibacterial activity of the formed CHX-nanogel complex before and after the surface functionalization of cationic polyelectrolyte was investigated [112]. It was found that compared with free CHX at the same concentration, this CHX-nanogel could greatly enhance the antibacterial effect of CHX.

The aerogel is a novel material formed by replacing the liquid of a gel with gas without changing its structure. A biopolymer-based aerogel has been developed to become one of the most potentially utilized materials in different biomedical applications. Alginate is a natural polymer mainly obtained from brown algae and seaweeds. It consists of linear copolymers of β-(1–4) linked d-mannuronic acid and β-(1–4)-linked l-guluronic acid units [113]. Trucillo et al. [114] entrapped the antibiotic ampicillin loaded liposomes in alginate aerogels by supercritical CO_2_ drying for prolonged drug release., and it was able to make the release of antibiotic from the aerogel double the normal time. The multi-free hydroxyl and carboxyl groups on the surface of alginate permit its cross-linking with several materials, which provides an outstanding candidate for antibacterial applications.

Thymol is a monoterpene phenolic derivative extracted from the antimicrobial plant. In Tohid Piri-Gharaghie et al.´s present study [115], thymol-loaded chitosan nanogels were prepared and their physicochemical properties were characterized. The encapsulation efficiency of thymol into chitosan and its stability were determined. The in vitro antimicrobial and anti-biofilm activities of thymol-loaded chitosan nanogel (Ty-CsNG), free thymol (Ty), and free chitosan nanogel (CsNG) were evaluated against both Gram-negative and Gram-positive multidrug-resistant (MDR) bacteria including *Staphylococcus aureus*, *Acinetobacter baumanii*, and *Pseudomonas aeruginosa* strains. Antibacterial activity tests revealed that Ty-CsNG reduced the MIC by 4–6 times in comparison to free thymol. In addition, the expression of biofilm-related genes including ompA, and pgaB were significantly down-regulated after treatment of strains with Ty-CsNG (*p* < 0.05). In addition, free CsNG displayed negligible cytotoxicity against HEK-293 normal cell lines and presented a new biocompatible nanoscale delivery system for enhancing antimicrobial and anti-biofilm activities.

Mao et al. [116] recently reported a mild pyrolysis approach to prepare carbon nanogels (CNGs) through polymerization and the partial carbonization of L-lysine hydrochloride at 270 °C as a potential broad-spectrum antimicrobial agent, which can inhibit biopolymer-producing bacteria and clinical drug-resistant isolates, and tackle drug resistance issues. CNGs possess superior bacteriostatic effects against drug-resistant bacteria compared to some commonly explored antibacterial nanomaterials (such as silver, copper oxide, and zinc oxide nanoparticles, and graphene oxide) through multiple antimicrobial mechanisms, including reactive oxygen species generation, membrane potential dissipation, and membrane function disruption. In addition, Petr Šálek et al. [117] successfully produced a well-defined and sub-micron 167 nm PDMAEMA-EDMA nanogel, which was later quaternized with iodomethane. This quaternized nanogel exhibited bactericidal activity against *Staphylococcus aureus* (*S. aureus*) and *Acinetobacter baumannii* (*A. baumannii*). The results illustrated that the quaternized PDMAEMA-EDMA nanogel acted as an effective bactericidal agent against both tested bacteria.

Nanogel delivery systems have been shown to be promising drug carriers, which play crucial role in the delivery of a wide variety of medicaments and therapeutics in bacterial and microbial diseases. In the future, parameters such as site specificity, selectivity, adverse effects, and therapeutic efficacy can be minimized by controlling the specific factors associated with formulation and delivery of nanogels, which results in the expanding horizon of nanogel delivery system in antimicrobial chemotherapy. More and broader in vivo and clinical research should be carried out for better cost-effective production of nanogel at commercial levels.

### 2.4. Nanoemulsion

Nanoemulsion (NE), also known as microemulsion, is a colloidal particulate nanosystem composed of a mixture of water, oil and a suitable stabilizing surfactant [118,119,120,121]. Nanoemulsions (NE) consist of nanometer-sized droplets stabilized by emulsifiers and are typically more stable and permeable due to their smaller particle sizes and higher surface area compared to conventional emulsions. The NE proved to be an effective drug delivery system due to their stability and solubility in aqueous media [122]. NE have been identified as a promising means of antimicrobial delivery due to their intrinsic antimicrobial properties, ability to increase drug solubility, stability, bioavailability, organ and cellular targeting potentials, capability of targeting biofilms, and potential to overcome antimicrobial resistance. NE can be administered through multiple different routes (oral, parenteral, dermal, transdermal, pulmonary, nasal, ocular, and rectal) [123].

Nanoemulsions have three different types of basic structures as shown in Figure 5. They mainly include: (1) (W/O) water-in-oil nanoemulsion, where the water phase is distributed in the oil phase, the inner phase is water, and the outer phase is oil. Surfactants and co-surfactants form a monomolecular film on the surface, and the (W/O) water-in-oil nanoemulsion, which coexists with the oil phase without resistance. (2) (O/W) oil-in-water nanoemulsion, where the inner phase is oil and the outer phase is water. The structure of (O/W) oil-in-water nanoemulsions is just opposite to that of (W/O) water-in-oil nanoemulsions. (O/W) oil-in-water nanoemulsions can coexist with water without resistance. (3) Bicontinuous nanoemulsion, part of which is surrounded by the water phase, where the droplets composed of the oil phase can be connected with the water droplets in the oil phase that together form the oil continuous phase and are surrounded by it, and the oil and water are constantly at the interface. The bicontinuous fluctuation between oil and water at the interface makes the bicontinuous nanoemulsion isotropic. Both internal and external phases of nanoemulsions can be loaded with drugs.

Coriander (*Coriandrum sativum*) oil was developed into a nanoemulgel by using a self-nanoemulsifying technique with Tween 80 and Span 80. Hydrogel material (Carbopol 940) was then incorporated into the nanoemulsion and mixed well [124]. Interesting results were obtained with the nanoemulgel against different types of bacteria, such as *Pseudomonas aeruginosa*, *Klebsiella pneumoniae*, and methicillin-resistant *Staphylococcus aureus* (MRSA), with a minimum inhibitory concentration (MIC) of 2.3, 3.75, and 6.5 μg/mL, respectively. The development of coriander oil into a nanoemulgel by using a self-nanoemulsifying technique showed a bioactive property better than that in crude oil. Thus, this simple nanotechnology techniques are a promising step in the preparation of pharmaceutical dosage forms.

Roya Moghimi et al. [125] measured the antibacterial activity of essential oils (Thymus daenensis) in pure and nanoemulsion form against the bacteria *Escherichia coli*. Antibacterial activity was determined by measuring minimum inhibitory concentration (MIC) and minimum bactericidal concentration (MBC); the mechanism of antibacterial activity was investigated by measuring potassium, protein and nucleic acid leakage in cells and electron microscopy. The results showed that when the essential oil (T. daenensis) was converted into a nanoemulsion, its antibacterial activity against an important food-borne pathogenic bacteria (*Escherichia coli*) was greatly enhanced. Based on the findings, the researchers propose that the mechanism of action for the antibacterial nanoemulsion against *E. coli* involves bringing the essential oil into close proximity with the cell membrane. This enables hydrophobic molecules in essential oils to disrupt cell membranes, possibly by altering the integrity of the phospholipid bilayer or by interfering with active transporters embedded in the phospholipid bilayer. Changes in the permeability of the disrupted cell membrane cause nucleic acids, proteins, and potassium to leak from inside the cell, resulting in cell death within 5 min. Therefore, nanoemulsions may be particularly effective delivery systems for essential oils due to their ability to facilitate antimicrobial application and increase antimicrobial efficacy.

Antimicrobial photodynamic inactivation (aPDI) is considered an effective approach to treat local microbial infections based on activation of photosensitizer (PS) by visible light, leading to overproduction of reactive oxygen species which inactivate pathogens and destroy cellular targets [126]. aPDI has been proven effective for eradication of a wide spectrum of human pathogens including bacteria (either Gram-positive Gram (+) or Gram-negative Gram (−)), fungi, protozoa, parasites, and viruses [127,128]. Zinc (II) phthalocyanine (ZnPc) is an efficient photosensitizer due to its low dark toxicity, strong absorption in the far red wavelength (>670 nm), long triplet life time, and high singlet oxygen quantum yield production [129,130]. Despite its favorable properties as a PS, its application is greatly limited by its poor water solubility, large molecular weight (MW  =  577.91), and highly lipophilic nature, causing it to be entrapped in the bilayer membranes of Gram (−) bacteria, and consequently hindering its transport to the target cytoplasmic membrane [126,131]. MahaFadel et al. [132] aimed to investigate the therapeutic effect of aPDI and nanoemulsion in combination for infections caused by drug-resistant bacterial strains. The results showed that ZnPc nanoemulsion improved antimicrobial photodynamic therapy in inactivating resistant bacterial infections and provided a promising therapeutic means of treating serious infections, and hence could be applied in diseases caused by other bacterial strains.

*Staphylococcus* is perhaps the pathogen of greatest concern because of its intrinsic virulence, ability to cause many life-threatening infections, its capacity to adapt to different environmental conditions, and biofilm formation [133,134]. Mennatallah A. Mohamed et al. [135] prepared novel nanobiotic formulations to improve the antibacterial activity of three antibiotics (linezolid, doxycycline, and clindamycin) against *Staphylococcus*. Antibiotics were formulated as nanoemulsions and evaluated for their antimicrobial activities and cytotoxicities. Results of this study revealed that antibiotics loaded in nanoemulsion had a higher antimicrobial activity and lower cytotoxicities as compared to those of conventional free antibiotics, indicating their potential therapeutic values.

Hassanshahian et al. [136] extracted the essential oil from the tropical *Alhagi maurorum* plant, and this essential oil was formulated as nanoemulsion using the ionotropic gelation method and chitosan as a nano-carrier. The effects of this nanoemulsion on the antibacterial, antibiofilm, and plasmid curing of six antibiotic-resistant pathogenic bacteria *(P. aeruginosa*, *E. coli*, *S. aureus*, *K. pneumonia*, *A. baumannii*, *B. cereus*) were evaluated. The results of antibacterial activity confirmed that this nanoemulsion had antibiofilm activity and better inhibition against bacteria compared to free essential oil. Moreover, this nanoemulsion had a good ability to cure and delete R plasmid. Zhao et al. [137] constructed a self-nanoemulsifying drug delivery system encapsulating buckwheat flavonoids, followed a pseudo-ternary phase diagram. The antimicrobial potential of this flavonoids nanoemulsion and flavonoids suspension were determined against *Staphylococcus aureus*, *Escherichia coli*, and *Candida albicans*. The results of MIC and minimal bactericidal concentration (MBC) determination exhibited that the antimicrobial activity of the nanoemulsions and suspension increased while enhancing the drug concentration, and the activity of nanoemulsion was significantly higher than that of the suspension against those three bacteria.

Self-emulsifying delivery systems (SNEDDS) are the isotropic mixtures of oils, surfactants, and co-surfactants that emulsify spontaneously upon mild agitation following the dilution with an aqueous media such as GI fluids that resulted with the formation of the nano scaled (<200 nm droplet size) oil-in-water type emulsion [138,139]. Delafloxacin (DFL) is a novel potent and broad-spectrum fluoroquinolone antibiotic agent effective against both Gram-positive and negative aerobic and anaerobic bacteria. Therapeutically, it is recommended for the treatment adult with infection of acute bacterial skin and skin structure infections (ABSSSIs) and community acquired pneumonia (CAP). Recently, to produce the optimal SNEDDS of the ethyl acetate fraction mangosteen peels, Pratiwi et al. [140], respectively, compared Tween 80, PEG 400 and Virgin Coconut Oil, and then analyzed the effectiveness of the optimal SNEDDS against *Staphylococcus* epidermidis. Anwer and coworkers [139] developed a DLF-encapsulated SNEDDS composed of Lauroglycol™-90, Tween^®^ 80 and Transcutol^®^-HP at different weight ratios by aqueous titration method. The Lauroglycol™-90 was selected based on its ability of enhancing the solubility and bioavailability of drug [141,142,143]. Furthermore, Tween-80 was used as a surfactant with a co-surfactant, Transcutol^®^-HP, as their mixture has been reported to improve the solubility and dissolution of various poorly soluble drugs from SNEDDS [144,145,146]. The optimized DLF-SNEDDS has demonstrated greater in vitro antimicrobial activity in respect to free DLF and levofloxacin.

Nanoemulsions may be particularly effective delivery systems for antibacterials due to their ability to facilitate antimicrobial application and increase antimicrobial efficacy. The findings of the current study may positively influence the pharmacodynamics of the antibiotics and consequently the dosage regimen of nanobiotics as well as on their clinical outcome.

### 2.5. Metal Nanoparticles

Inorganic NPs include metal and metal oxides, such as gold (Au), silver (Ag), platinum (Pt), iron (Fe), iron oxide (Fe_3_O_4_), titanium oxide (TiO_2_), copper oxide (CuO), and oxide Zinc (ZnO) [147,148]. For a long time, inorganic NPs have been studied for various therapeutic applications, such as anticancer and antibacterial treatment. Similar to organic drugs, inorganic drugs can also benefit from NDDSs by improving their pharmacokinetic properties, such as enhanced targeting, drug loading, and immune system evasion [149]. The antibacterial effect of metal NPs depends on the shape and size, and it is performed through several modes of action, including the ability to bind and change the function of polymers and produce free radicals through reactive oxygen species (ROS) [150,151]. Metal NPs are considered as valuable candidates for antimicrobial agents by targeting different cell components, such as cell wall, DNA, membrane and protein [152].

Green synthesized metallic and metal oxide NPs are considered as the potential means to target bacteria as an alternative to antibiotics. Nanoconjugates have also attracted attention because of their increased biological activity as compared to free antibiotics. In Ghaffar et al.´s investigation [153], AgNPs, ZnO NPs, CuO NPs, and FeO NPs have been synthesized by using leaf extract of *Ricinus communis*. This study shows that the *S. aureus* resistant to streptomycin becomes susceptible to the same antibiotic in combination with nanoparticles. Synergistic antibacterial properties of nanoparticles with antibiotics provide an alternative approach to minimize drug resistance and provide potential applications in the medical field. Thus, biogenic metallic and metal oxide nanoparticles may be also used effectively in combating Multidrug-resistant strain (MDRS).

Magnetic NPs have received widespread use in the field of biomedical and nanomedicine [154]. The antibacterial activity of this class of NPs are intriguing because of their damaging effect to the bacteria through interfering with the thiol group [155]. The antibacterial efficacy of biosynthesized magnetic NPs was tested against various drug-resistant bacteria, such as *E. coli*, *Shigella*, *P*. *aeruginosa*, *S. aureus*, *Salmonella typhi*, and *Pasteurella multocida*. Agar-diffusion method confirmed the efficiency of magnetic NPs to suppress the growth of *S. aureus* and *E. coli* in a concentration-dependent manner. Moreover, some magnetic NPs displayed strong efficacy against all bacteria when compared with the standard drugs [156]. Recently, Zargarnezhad et al.´s study [157] demonstrated that the enzyme-mimetic activities of magnetic nanoparticles could accelerate the activation process of isoniazid against mycobacterial and non-mycobacterial microorganisms (e.g., *Enterococcus faecalis*, *Escherichia coli*, *Pseudomonas aeruginosa*, and *Staphylococcus aureus*.) as well. 

AuNPs do not exert (inherent) antibacterial activity; however, they enhance antibacterial properties of the loaded antibacterial drugs [158]. Drug-loaded AuNPs could collapse bacteria membrane potential, hinder ATPase activity, and halt binding of ribosomes to tRNA leading to deteriorated bacterial cell metabolism [159]. Antibiotic ciprofloxacin (CIP) is widely used for the treatment of numerous bacterial infections in joints, bones, skin, tooth, gastrointestinal, and urinary and respiratory tracts [160]. Recently, Ciprofloxacin-loaded gold nanoparticles (CIP-AuNPs) were prepared [161]. These CIP-AuNPs were stable and exerted enhanced in vitro antibacterial activity against *E. faecal*, compared with free CIP. Moreover, unlike CIP, CIP-AuNPs were non-hemolytic. 

AgNp have highly strong antimicrobial activity for multidrug resistant microorganisms [162]. The AgNPs-chitosan/montmorillonite nanocomposite films may control the morphological characteristics of AgNPs besides the significant inhibition of *E. coli* and B. subtilis [36]. Tripolyphosphate (TPP) was added to the chitosan film as a crosslinker and a reducing agent of the silver nanoparticles agent, while Chitosan(CS)-based NPs have shown a tremendous potential as an antibacterial agent [163,164,165,166,167,168,169,170,171]. CS/TPP-AgNPs film inhibited *S. aureus* and *E. coli* more efficiently compared to the effect of chitosan films alone [172]. The silver nanoparticle-loaded microspheres (ChM-Ag) was very efficient in inhibiting *E. coli* and *S. aureus* [173]. An eco-friendly CS-AgNPs hybrid was developed from AgNPs biologically prepared using *T. portulacifolium* leaf extract as a reducing agent, and the inhibitory effects of these hybrid NPs were tested against two bacterial strains: *E. coli* and *Serratia marcescens*. These hybrid CS-AgNPs inhibited the growth of *E. coli* and S. marcescens. The antibacterial activity of CS-AgNPs increased with the increase in the concentration of CS-AgNPs [174]. Recently, the anti-biofilm activity of CS-AgNPs was tested against pathogenic bacteria including *S. aureus* and *P. aeruginosa*. CS-AgNPs inhibited the growth of *S. aureus* (85%) and *P. aeruginosa* (95%) at 100 μg/mL [175].

Gold nanoparticles capped with chitosan (CS-AuNPs), glycol-chitosan (GC-AuNPs), and poly-γ-glutamic acid (PA-AuNPs) were applied on fabrics and evaluated for antibacterial activity. PA-AuNPs showed a higher antibacterial activity against *S. enterica* compared to gentamycin, while CS-AuNPs and GC-AuNPs exhibited maximum antibacterial activity against Listeria monocytogenes, followed by *S. enterica*, *E. coli*-O157:H7, and *S. aureus* [176]. Besides, Titania nanocomposites (PEG/TiO2NPs), and pseudopolyrotaxane-titania-nanocomposite (PEG/α-CD/TiO2NPs) have been synthesized as novel alternative antibiotic strategies against antibiotic-resistant microorganisms [177]. Subsequently, these NPs were used as drug carriers for sulfaguanidine. 

A hydrophobic structure of conjugated polymers(CPs) enables the development of conjugated polymer nanoparticles (CPNs) in water, allowing nanoparticles to achieve further function by modifying surfaces and their core [178]. Due to its high absorption potential, conjugated nanopolymers (CNPs) can sensitize the environment’s oxygen to create cytotoxic reactive oxygen species (ROS) under light irradiation [179]. The ROS break the microbial membrane and degrade it with the help of photosensitizers [180]. For instance, Nanoarchitectonically constructed conjugated nanopolymerpoly complexing with photosensitizers could kill more than 95% *E.coli* [181].

### 2.6. Solid Lipid Nanoparticles

Solid lipid nanoparticles (SLNs) are nanoparticles comprised of solid lipids stabilized by an emulsifying layer in aqueous dispersion. They are similar to nanoemulsions in which the inner liquid lipid is replaced with a solid lipid [182]. These are a type of lipid-based vesicular structure that has received popularity because of their capacity to deliver drugs at a controlled and site-specific rate [183]. Additionally, surfactants are employed in emulsification to improve SLN stability [184]. They are biopolymers of natural or synthetic origin and are suitable for encapsulating lipophilic drugs [185]. SLNs containing antitubercular prepared using microemulsion technique drugs showed twofold inhibition of *Mycobacterium marinum* compared to pure antitubercular drugs [186]. Jalal et al. [187] reported an improvement in antibacterial function of rifampicin (Rif)-loaded SLNs, against *B. abortus*, higher than unbound rifampicin by two folds. To avoid the side effects of Vancomycin (VAN), including renal failure and blood disorders such as neutropenia [188,189,190], Seedat et al. demonstrated that the co-encapsulation of multi-lipids and polymers could enhance the performance of VAN in lipid polymer hybrid nanoparticles (LPNs) by improving its entrapment as well as its release profile and antibacterial activity against both sensitive and resistant bacterial strains [191]. 

Multiple lipid nanoparticles (MLNs) systems are the third generation of lipid nanoparticles that have emerged as alternative to nanostructured lipid carriers to overcome some of their shortcomings [192,193]. MLNs, combining the advantages of double emulsions and solid lipid nanoparticles, allow the simultaneous encapsulation of hydrophilic and lipophilic compounds guarantying a combined therapy against many diseases [193]. The literature survey showed that Carvacrol (CAR, an antimicrobial natural phenolic monoterpene [194,195,196,197,198])-loaded nanoparticles have more attractive characteristics, such as good chemical stability, high solubility, low toxicity, controlled release rate as well as enhanced bioavailability [199,200,201]. Khalifa et al. prepared the MLNs containing CAR and VAN co-loaded (CAR-VAN-MLNs) by the ultra-sonication technique and investigated for both physico-chemical and antimicrobial properties. Antimicrobial studies suggested that the co-encapsulation improved VAN antibiotic activity against *Staphylococcus aureus* with a possible greater advantage during systemic therapies [202].

Lipid polymer hybrid nanoparticles (L-P-NPs), which possess both lipid and polymeric carriers, were synthesized to overcome the limitations of both liposomes and polymeric nanoparticles. L-P-NPs are effective in encapsulating the hydrophobic molecules with a higher drug payload than biopolymer-based nanoparticles due to their nano-range size and large surface areas. In addition, they improve drug stability and have the ability to improve the oral bioavailability of poorly water-soluble drugs [203]. Recently, lipid-chitosan hybrid nanoparticles (LIPOMER nanoparticles) have been used for the oral delivery of some poorly soluble drugs [204,205,206,207]. In addition, the role of L-P-NPs in encapsulating antibiotics, e.g., chitosan-coated lipid nanoparticles loaded with rifampicin, have been evaluated for the better management of tuberculosis [208,209]. Anwer et al. [210] developed novel DFL-loaded stearic acid (lipid) chitosan (polymer) hybrid nanoparticles (L-P-NPs). The results showed that the optimized DFL-loaded L-P-NPs were more potent against both Gram-positive and negative strains of bacteria and highly bioavailable in comparison to delafloxacin normal suspension.

SLNs can not only serve as antibiotic drug delivery systems, but some nanoparticles themselves have antibacterial effects. Using nanoparticles as a nano-drug delivery system for antibiotics can improve the antibacterial effect of antibiotics. While it is difficult for antibiotics to penetrate human cells, nanoparticles, due to their small size and structure, can penetrate the membranes of affected cells. Because of this, they play a role in fighting intracellular pathogens such as multidrug-resistant *Staphylococcus* (*MRSA*). Nanoparticles can also be used in combination with polymers or liposomes to carry antibiotics for more effective antibacterial effects.

## 3. Conclusions

The invention and application of antibacterial drugs are some of the greatest achievements in the field of medicine in the 20th century, effectively curing infectious diseases caused by many types of pathogens and reducing related mortality. However, with the widespread use and even abuse of antibacterial drugs, the problem of antimicrobial resistance has become more and more serious in recent years, making commonly used antibacterial drugs ineffective or even completely inactive, posing a major threat to human health. Numerous studies have shown that NDDSs, including liposomes, polymeric micelles, etc., can improve the pharmacological and pharmacological properties of parent drugs by extending the duration of action, improving efficacy and targeting, overcoming drug resistance, reducing immunogenicity and toxicity. Several NDDSs have their own advantages and disadvantages (Table 1).

In this context, NDDSs are regarded as one of the means to solve this problem, and this strategy has proven to be effective. In recent years, the nenomedicine drug delivery systems have attracted the interest of many researchers, and many studies have focused on the study of the antibacterial drug nenomedicine drug delivery system, by using the system to enhance the targeting and permeability of antibacterial drugs. At present, several liposome preparations have been listed or are in clinical trials for infectious diseases. For example, Arikayce liposome drugs for the treatment of Mycobacterium avium have been approved for clinical use by FDA; Paclitaxel Polymeric Micelles for Injection for cancer treatment have been approved in China, which shows their therapeutic potential and the possibility of rapid approval of subsequent formulations. Therefore, NDDSs are regarded as an effective strategy to address antibiotic resistance. Among them, liposomes are currently the most mature nano-drug carriers. However, NDDSs still have a long way to go, and much research in this area still waits to be carried out. Moreover, there is still a lack of a large number of relevant in vivo studies on antibiotics loaded by nano-drug delivery systems. Judging from the marketed nano-carrier drugs, these nano-drug delivery systems are currently mainly used in the field of cancer treatment, so there is still much room for development in the field of antibiotic delivery. It was believed that in the next few years, more NDDSs with enhanced potency will become clinically available and become effective tools for overcoming antimicrobial resistance. In conclusion, nanotechnology can lead to a breakthrough in the development of various nanoantimicrobial agents, which may reduce the public health threats from recalcitrant infectious diseases and biofilm-associated infections.

## Figures and Tables

**Figure 1 nanomaterials-12-01855-f001:**
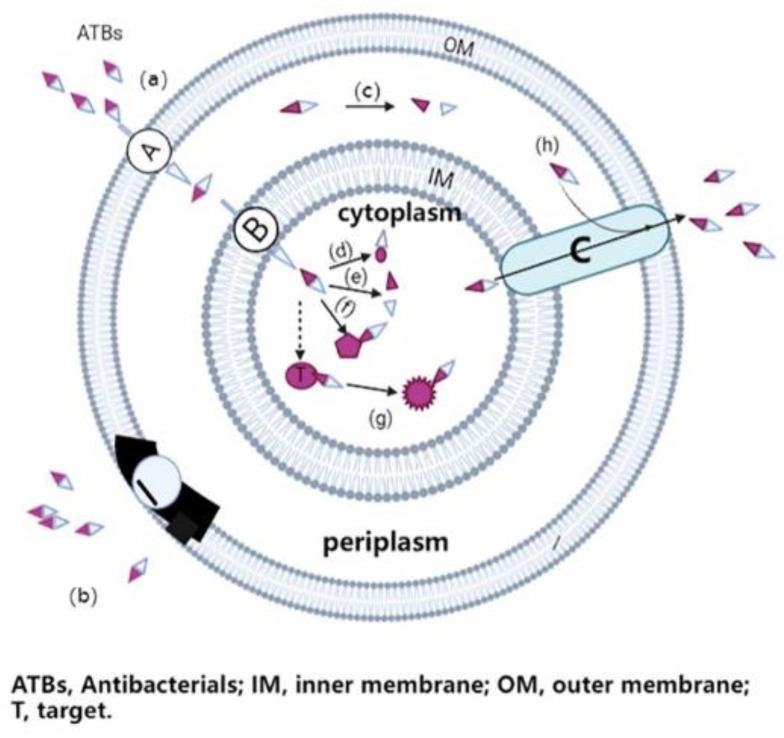
The main resistance mechanisms of gram-negative bacteria to antibiotics.

**Figure 2 nanomaterials-12-01855-f002:**
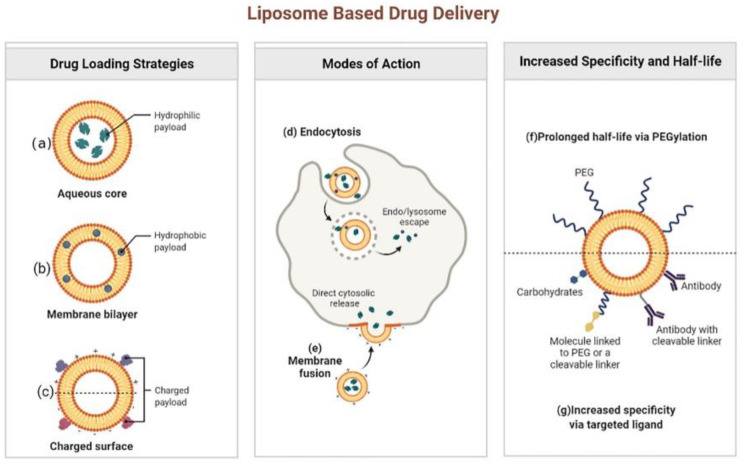
Liposome based drug delivery. Reprinted from ref. [28]. Copyright 2022 by BioRender.

**Figure 3 nanomaterials-12-01855-f003:**
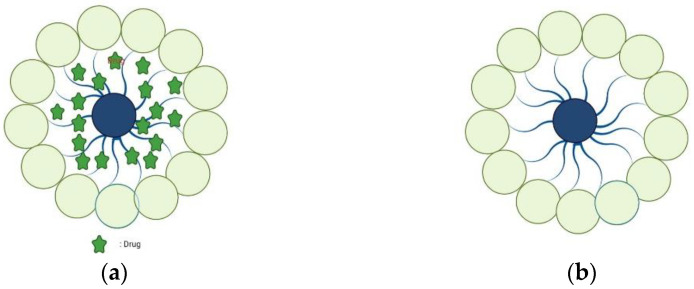
Blank polymer micelles (**a**) and drug-loaded polymer micelles (**b**).

**Figure 4 nanomaterials-12-01855-f004:**
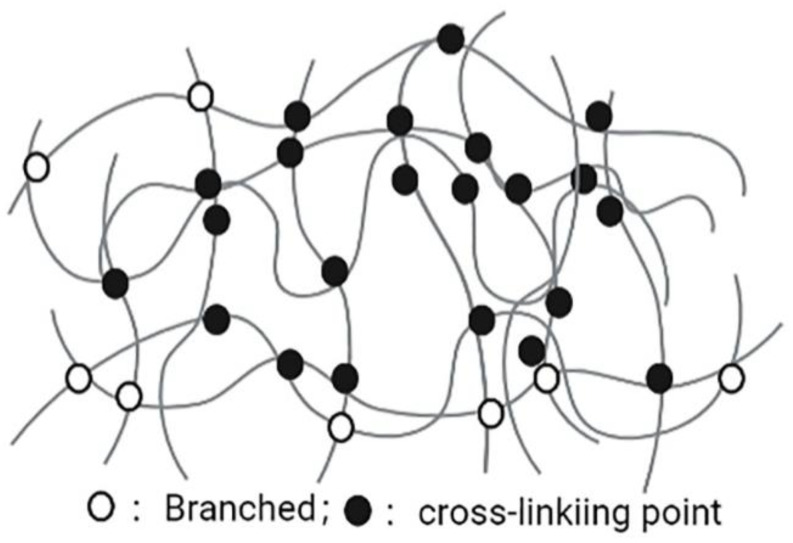
Nanogel.

**Figure 5 nanomaterials-12-01855-f005:**
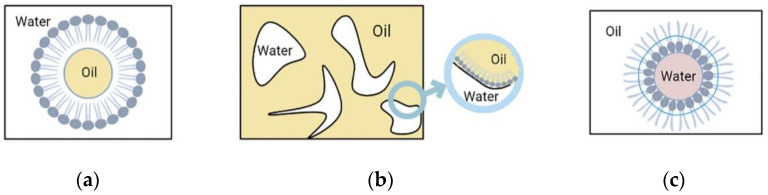
Nanoemulsion.

**Table 1 nanomaterials-12-01855-t001:** A brief summary of several NDDSs.

NDDS	Size	Advantage	Disadvantage
Liposomes	25–1000 nm	It can encapsulate both water-soluble drugs and lipid-soluble drugs; it can be modified; enhance drug targeting; prolong drug action time and improve drug stability; help overcome multidrug resistance; reduce adverse drug reactions etc.	It is unstable and easy to be hydrolyzed; it is prone to auto-oxidation, resulting in reduced membrane fluidity, drug seepage, and toxicity after aggregation and precipitation.
Polymeric micelles	10–100 nm	First, Polymeric micelles have lower critical micelle concentration, larger compatibilization space, stable structure, and can encapsulate drugs by chemical, physical, and electrostatic methods according to the different properties of the hydrophobic segment of the polymer; secondly, the drug is encapsulated by nanoparticles, which avoids the allergic reaction of natural drugs and does not require pretreatment. Third, the dosage of Active Pharmaceutical Ingredient required can be greatly increased through the supplementation of nanomicelles.	The stability of polymer micelles will be greatly weakened after entering the human body, resulting in premature release of drugs and loss of targeting effect. At present, there are many means to increase the stability of micelles, including cross-linking the hydrophobic core, adding conjugated targeting molecules at the end of the segment, etc. However, as the design of micelles becomes more and more complex, the evaluation of its performance is also difficult. Correspondingly, it becomes difficult, time-consuming and labor-intensive, which is not conducive to the clinical application of polymer micelles.
Nanogels	<200 nm	First, they are small in size and easy to be phagocytosed by cells; second, they can easily penetrate various protective membranes in the human body, such as the meninges, so that drugs can be administered to the brain; third, the drug-carrying efficiency is high.	Its preparation requires the participation of organic solvent.
Nanoemulsion	<200 nm	It can make the oil phase and the water phase together; increase the solubility of the drug, improve the bioavailability of the drug, and avoid the first-pass effect of the liver.	Its preparation cost is relatively high. Most of the nanoemulsion processes are prepared by the emulsification of a high-speed emulsifier. The investment in equipment and the technological requirements for the emulsification process are relatively high. In addition, nanoemulsion is a liquid preparations less stable than solid preparations. The shelf life of nanoemulsion drugs is usually 6 to 18 months, while that of solid preparations is 2 to 3 years. Many liquid preparations of traditional Chinese medicine cannot use nanoemulsion technology.
Nanoparticles	<100 nm	It can probe into the cell wall of the pathogenic microbes and even have the capacity to intrude into cellular pathways. Nanoparticles themselves are capable of destroying unwanted foreign particles or toxic cells, which enter into our bodies. Nanoparticles can combine with specific drugs and deliver to target specific cells with lesser side effects. Nanoparticles showed higher internalization uptake, accumulation and retention time of drug improving antibacterial activity drug decreasing an-timicrobial resistance and also inhibit biofilm formation	Both naturally derived nanoparticles and synthetic nanoparticles are toxic, which also hinders the progress of clinical applications.

## Data Availability

Not applicable.

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
