# Peer review of "Recent Advances in Antimicrobial Nano-Drug Delivery Systems"

_nanomaterials, 2022, doi:10.3390/nano12111855_

Round 1

Reviewer 1 Report

The manuscript “Recent advances in antimicrobial nano-drug delivery systems” is a review on the use of nanotechnology to improve bioavailability and targeting of antimicrobial drug systems. This is a very accurate and well organized work, on a hot topic. Therefore, it deserves publication, but after some minor revisions, as follows:

- Add a new paragraph on the use of emerging production techniques (such as supercritical CO2 assisted processes, high-pressure, microfluidic, etc.) to obtain these nanodevices. Read for instance: 10.1016/j.jcou.2021.101669, 10.3390/antibiotics9100648, 10.1016/j.jcou.2020.101161, 10.1016/j.ijpharm.2021.121403, etc..

- Use italic for microorganisms.

- Correct typos.

Author Response

Comments:

The manuscript “Recent advances in antimicrobial nano-drug delivery systems” is a review on the use of nanotechnology to improve bioavailability and targeting of antimicrobial drug systems. This is a very accurate and well organized work, on a hot topic. Therefore, it deserves publication, but after some minor revisions, as follows:

Response: We are grateful for the reviewer’s nice judgement.

Suggestions:

  • Add a new paragraph on the use of emerging production techniques (such as supercritical CO2 assisted processes, high-pressure, microfluidic, etc.) to obtain these nanodevices. Read for instance: 10.1016/j.jcou.2021.101669, 10.3390/antibiotics9100648, 10.1016/j.jcou.2020.101161, 10.1016/j.ijpharm.2021.121403, etc..

Response: Dear reviewer#1, Thank you so much for your comment.

We have chosen to focus on drug deliveries and the nanoparticles that we have experience with, and some new NDDS production techniques have been cited in the revised manuscript. However, the four suggested references listed above are not cited because we think that these would not fit in the scope of this review.

  • Use italic for microorganisms. And correct typos.

Response: As for the minor errors, it was done.

Reviewer 2 Report

The title is „Recent advances in antimicrobial nano-drug delivery systems” but in my opinion the paper does not represent recent scientific developments in the field. The publication does not include recent reports from this year, for example, typing "antimicrobial nanoparticles" into the Web of Science database shows about 740 records for 2022 year and more than 4,000 for 2021, many of which are related to the topic of the presented publication. The publication with 100 citations contains 6 citations from 2020 and 6 from 2021. What more, reviews on a similar topic have appeared in recent years : „: Osman N. at all. Surface modification of nano-drug delivery systems for enhancing antibiotic delivery and activity. Wiley Interdisciplinary Reviews: Nanomedicine and Nanobiotechnology, 14(1), (2022) e1758”, „Shabnam S., at all, Nanoparticles as antimicrobial and antiviral agents: A literature-based perspective study, Heliyon 7 (2021) e06456”. In summary, in my opinion, the publication would be valuable if it described the latest scientific developments that would indicate the great potential for the use of nanomaterials in antimicrobial therapy or vice versa. Therefore, I believe that the presented materials in the paper are not suitable for printing without the major revision because they do not contribute anything significant to the recent science and because similar studies are available in the literature.

Author Response

Comments:

The title is Recent advances in antimicrobial nano-drug delivery systems” but in my opinion the paper does not represent recent scientific developments in the field. The publication does not include recent reports from this year, for example, typing "antimicrobial nanoparticles" into the Web of Science database shows about 740 records for 2022 year and more than 4,000 for 2021, many of which are related to the topic of the presented publication. The publication with 100 citations contains 6 citations from 2020 and 6 from 2021. What more, reviews on a similar topic have appeared in recent years : „: Osman N. at all. Surface modification of nano-drug delivery systems for enhancing antibiotic delivery and activity. Wiley Interdisciplinary Reviews: Nanomedicine and Nanobiotechnology, 14(1), (2022) e1758”, „Shabnam S., at all, Nanoparticles as antimicrobial and antiviral agents: A literature-based perspective study, Heliyon 7 (2021) e06456”. In summary, in my opinion, the publication would be valuable if it described the latest scientific developments that would indicate the great potential for the use of nanomaterials in antimicrobial therapy or vice versa. Therefore, I believe that the presented materials in the paper are not suitable for printing without the major revision because they do not contribute anything significant to the recent science and because similar studies are available in the literature.

Response: We highly thank you for sharing your perspective, which we take as a very pertinent suggestion.

In the revised manuscript, more important scientific work published in recent two years have been summarized in the revised manuscript. Thus, the text was reformulated and new references  were added in order to better adapt to the objective of the work.

Reviewer 3 Report

Review of Nanomaterials-1659754, “Recent advances in antimicrobial nano-drug delivery systems”, by Tong-Xin Zong, Juan Zhang, Cheng-Shi Jiang, Luis Alexandre Muehlmann and Shan-Kui Liu.

My ability to review this manuscript rests on the fact that my students and I do research on nanoparticulate drug delivery systems. My view is that this manuscript is not worthy of publication. I base this view on two points: (1) the review is far too limited and (2) it was carelessly put together.

Concerning the first point, I ask myself at which cohort this review is aimed. Each subsection is about 1½ pages long and touches only lightly on the subject. Further, important subjects, such as metal nanoparticles, metal oxide nanoparticles, etc., are omitted. In comparison, I just came across a new review on magnetic nanostructures (Chem. Rev., 2022, 122, 5411−5475), a subject that the present review does not consider, which is 65 pages long. While I do not advocate 65 pages per subsection, something longer than 1½ pages would be appropriate. I do, however, advocate something more that four subsections, especially when the manuscript title implies a complete overview.

Concerning the second point, I get the impression that each subsection was written by a different person, with little or no attempt to blend them. Consider, for example, the pronoun, “I”, on line 174; why not, “We”? Consider also line 371, which bears the same subsection number and title as line 261.

Based on my view, I recommend that this manuscript be rejected.

Author Response

Comments:

Review of Nanomaterials-1659754, “Recent advances in antimicrobial nano-drug delivery systems”, by Tong-Xin Zong, Juan Zhang, Cheng-Shi Jiang, Luis Alexandre Muehlmann and Shan-Kui Liu.

My ability to review this manuscript rests on the fact that my students and I do research on nanoparticulate drug delivery systems. My view is that this manuscript is not worthy of publication. I base this view on two points: (1) the review is far too limited and (2) it was carelessly put together.

Concerning the first point, I ask myself at which cohort this review is aimed. Each subsection is about 1½ pages long and touches only lightly on the subject. Further, important subjects, such as metal nanoparticles, metal oxide nanoparticles, etc., are omitted. In comparison, I just came across a new review on magnetic nanostructures (Chem. Rev., 2022, 122, 5411−5475), a subject that the present review does not consider, which is 65 pages long. While I do not advocate 65 pages per subsection, something longer than 1½ pages would be appropriate. I do, however, advocate something more that four subsections, especially when the manuscript title implies a complete overview.

Concerning the second point, I get the impression that each subsection was written by a different person, with little or no attempt to blend them. Consider, for example, the pronoun, “I”, on line 174; why not, “We”? Consider also line 371, which bears the same subsection number and title as line 261.

Based on my view, I recommend that this manuscript be rejected.

Response: We thank the reviewer#3 for evaluating our manuscript and for the very pertinent comments.

We agree that the manuscript presented alignment errors between the subtopics, however, after an intense review, we reformulated the writing of the work.

As for the number of pages per subtopic, we tried to summarize the objectivity and importance of each nanoparticle, given that it would be possible to carry out an intense review for each one. In the revised manuscript, more important works (such as metal nanoparticles) published recently has been summarized, and 210 references were cited to make the paper being acceptable 26 pages.

Thank you again for the instructive comments  

Reviewer 4 Report

overall this paper has well-constructed, but some important points have to be clarified or fixed before we can proceed and positive action can be taken

Abstract: Please focus on the abstract. In particular, the last two sentences are very general. I would prefer to see recent particular advances in antimicrobial drug delivery nanosystems which makes them important in the abstract, rather than a description of general description. 

The author has only reviewed some nano-drug delivery systems, although they are many recent advances took place in other nano-drug delivery systems. I would prefer to include some concepts of nano-drug delivery systems in their recent publications to be cited for example

https://doi.org/10.1016/j.molstruc.2021.130844 etc. ...

Conclusion: In the conclusion of the manuscript, I could find some mistakes in typing and prefer it to be more descriptive.

Author Response

Comments:

overall this paper has well-constructed, but some important points have to be clarified or fixed before we can proceed and positive action can be taken

Response: We are grateful for the reviewer’s positive judgement and sharing suggestion.

  • Abstract: Please focus on the abstract. In particular, the last two sentences are very general. I would prefer to see recent particular advances in antimicrobial drug delivery nanosystems which makes them important in the abstract, rather than a description of general description.

Response: We reformulated the abstract as you suggested.

  • The author has only reviewed some nano-drug delivery systems, although they are many recent advances took place in other nano-drug delivery systems. I would prefer to include some concepts of nano-drug delivery systems in their recent publications to be cited for example https://doi.org/10.1016/j.molstruc.2021.130844 etc. ...

Response: Many thanks for the meaningful instruction. The reference suggested by reviewer#4 have been cited, and also more recent advances associated with potential for clinical use as drug-resistant antibacterial have been cited in the revised manuscript.

  • Conclusion: In the conclusion of the manuscript, I could find some mistakes in typing and prefer it to be more descriptive.

Response: The manuscript has been checked very careful, and the typos have been corrected.

Round 2

Reviewer 2 Report

It is OK.

Reviewer 3 Report

The manuscript is much improved, and more care has obviously been taken in its organization. I have no fundamental objections to its being accepted for publication. However, it adds little to our knowledge, and is only one of many such reviews presently being published.